# An Automotive LiDAR Performance Test Method in Dynamic Driving Conditions

**DOI:** 10.3390/s23083892

**Published:** 2023-04-11

**Authors:** Jewoo Park, Jihyuk Cho, Seungjoo Lee, Seokhwan Bak, Yonghwi Kim

**Affiliations:** 1Durability Technology Team, Hyundai Motor Company, Hwaseong 18280, Republic of Korea; 2IT Convergence Components Research Center, Korea Electronics Technology Institute (KETI), Gwangju 61005, Republic of Korea

**Keywords:** LiDAR, LiDAR performance test, sensor validation, real road sensor test

## Abstract

The Light Detection and Ranging (LiDAR) sensor has become essential to achieving a high level of autonomous driving functions, as well as a standard Advanced Driver Assistance System (ADAS). LiDAR capabilities and signal repeatabilities under extreme weather conditions are of utmost concern in terms of the redundancy design of automotive sensor systems. In this paper, we demonstrate a performance test method for automotive LiDAR sensors that can be utilized in dynamic test scenarios. In order to measure the performance of a LiDAR sensor in a dynamic test scenario, we propose a spatio-temporal point segmentation algorithm that can separate a LiDAR signal of moving reference targets (car, square target, etc.), using an unsupervised clustering method. An automotive-graded LiDAR sensor is evaluated in four harsh environmental simulations, based on time-series environmental data of real road fleets in the USA, and four vehicle-level tests with dynamic test cases are conducted. Our test results showed that the performance of LiDAR sensors may be degraded, due to several environmental factors, such as sunlight, reflectivity of an object, cover contamination, and so on.

## 1. Introduction

The Light Detection and Ranging (LiDAR) sensor is a transformative technology that detects the surrounding environment of an autonomous car driving in real-time. Thanks to their ability to measure long-range spatial information in a few microseconds, LiDAR sensors have become widely used in level three, or higher levels, in driving automated cars [1], as well as in standard Advanced Driver Assistance Systems (ADASs). The functional safety and reliability of such driving assistance are highly correlated with the quality of perception of the automotive sensors (Front camera, Radar, and LiDAR), which have different mounting positions, redundancy designs, and intrinsic characteristics [2]. Due to such variations, direct quality assessment for automotive sensors remains an open research topic.

The most standard protocols for ADAS & self-driving functional safety focus on system-level testing, not sensor-level testing [3]. The quantitative evaluation of LiDAR’s performance under harsh environments, such as fog [4], rain [5], background light [6], dust [7], and snow [8], is essential for providing safe ADAS and self-driving features, for both car consumers and manufacturers, in terms of proper redundancy design. It is an ongoing research field to expand test cases based on the operation design domain (ODD) of automotive cars.

Many reports in the literature refer to the performance deterioration of LiDAR sensors under harsh environments over decades. Yet, LiDAR test methods are not standardized for several reasons. First, qualitative measurement and reproduction of environmental factors are extremely difficult (e.g., visibility of fog). Second, the LiDAR system itself has unique characteristics in 3D measurement and in having different form factors (wavelengths of laser, beam steering mechanisms, or photon transmitting and receptive modules).

The existing LiDAR validation methods examine its capability and signal repeatability in an indoor facility [9], a static scenario [10], and virtual scene [5] by measuring the rate of performance drop with a controllable environmental factor. In these cases, a quantified environmental factor is controlled in a test facility, while a LiDAR sensor captures signals received from a referent target. Recently, related work deals with vehicle-level validation in laboratory conditions [4,11]. Considering the fact that actual ODD of the automotive car is more complex than the artificial environments, it is still necessary to examine automotive LiDAR performance in a real driving fleet under different weather conditions. In our review, the state-of-the-art methods investigate LiDAR capabilities in real driving cases [8,12]. Still, their results are subjected to a qualitative evaluation by means of human vision, such as visual comparison of LiDAR scan data, pass & fail tests judged by experts, and laborious work on statistics.

In this paper, we propose a LiDAR performance test method that fully accounts for quantitative analysis in real road driving. We propose an unsupervised learning method for feature extraction of reference targets from point clouds of a LiDAR sensor, which can be utilized to produce quantitative data statistics for LiDAR performance tests under dynamic situations. For LiDAR performance evaluation, we demonstrate eight different environmental tests, including a dynamic case in which both the ego-vehicle and counter-vehicle are driving through a real road at a constant speed. Our environmental tests were twofold. First, we simulated the performance degradation of an automotive LiDAR sensor under the following adverse cases: high sunlight, high ambient temperature, low ambient temperature, and cover contamination by dust. Our simulations were based on time-series captures of ambient temperature, sunlight, and visibility against dust contamination from real road fleets in Nevada and Alaska, the USA. Second, we also discovered variations of LiDAR signal characteristics for four driving scenes: daylight versus night, color of front vehicle, vibration on a country road, and interference by LiDAR by an oncoming vehicle. We performed real road tests for these cases with our feature extraction algorithm.

In summary, our contributions are as follows:We show a vehicle-level performance evaluation pipeline consisting of a schematic diagram of the test vehicle, the test metric & procedure, and the environmental factors.We present performance test results for an automotive-graded LiDAR sensor, in both a real road driving scenario and an environmental simulation based on time-series measurements from real road fleets in harsh weather conditions.We propose a spatio-temporal point segmentation algorithm using the density-based unsupervised clustering algorithm that can be utilized in dynamic test scenarios (e.g., feature extraction from moving objects, such as a car).

## 2. Related Work

Literature on LiDAR performance validation in adverse conditions can be categorized into three groups. The first group focuses on performance degeneration in a weather simulation using a rain/fog chamber. The second group involves sensor validation in a dynamic scene where a test vehicle, or target, moves, based on test scenarios. The third group expands on the test cases using physical simulations in combination with high-fidelity modeling of sensors, the surrounding objects, and the environment. In this section, we briefly introduce existing studies related to each group.

### 2.1. Lidar Performance Evaluation in Adverse Weather Conditions

Many studies have been conducted to evaluate the performance degradation of automotive LiDAR in adverse weather conditions. Wojtanowski et al. [13] analyzed the range degradation of 1550 nm and 905 nm LiDAR in rain and fog conditions using a numerical analysis of signal-to-noise (SNR) correlated to the laser wavelength. Instead of a laboratory test and numeric simulation, Kutila et al. [14] performed a relative comparison of 1550 nm and 905 nm LiDAR sensors’ performances in fog and rain conditions, as well as conducting a vehicle-level test for commercial 905 nm LiDAR sensors in a rain/fog chamber. Rasshofer et al. [10] and Filgueira et al. [15] quantified the influence of rain on Time-of-flight (ToF) LiDAR through empirical tests in outdoor scenes, finding that LiDAR capabilities varied, based on the amount of rain and object material. Similarly, Hasirlioglu et al. [9] investigated the influence of fog on automotive LiDAR using an indoor fog simulator. Li et al. [4] modeled a range process of ToF LiDAR under artificial fog conditions with visibility recording data against fog density, providing a Gaussian process to estimate the operational feasibility of a LiDAR under known fog density. Caballo et al. [16] introduced a benchmark dataset for LiDAR sensors, which includes static chamber test data under adverse weather conditions. In particular, they captured point cloud degradation caused by fog, rain, and strong sunlight for six commercial LiDAR sensors.

The early stage test setup in the aforementioned work was limited to a sensor-level test in which the test LiDAR was installed towards a static target. For an automotive LiDAR sensor, quality assessment in a variety of dynamic scenes must be included, in order to account for sensor errors possibly occurring in the ODD of the automotive vehicle which is to be combined with the LiDAR sensor.

### 2.2. Lidar Performance Test in Dynamic Scene

The previous work in this category aimed to provide dynamic tests for the automotive LiDAR. Dannheim et al. [17] proposed a weather detection system using LiDAR scan data in driving situations. Their results were validated in laboratory simulations. Tang et al. [18] performed a driving test for autonomous LiDAR sensors under rainy conditions, covering pedestrian detection in sunny and rainy weather using system-level metrics, such as stop distance, time to stop, hard brake, and LiDAR/camera detection. Heinzler et al. [5] developed test scenarios that are more realistic in both driving and static conditions. In their experimental setup, pedestrian dummies, real traffic signs, bicycles, and a driving car were placed and moved against a test vehicle in a fog chamber. However, the dynamic test results, which mainly focused on the effect of multi-echo versus different fog density, were restricted to visual comparison by human observers. Kim et al. [11] verified the number of points received from a static target of different materials (wood, plastic, etc.) according to the driving speed of a test vehicle and the amount of rainfall, using a rain simulation chamber. These methods offer LiDAR capabilities against different weather conditions in dynamic scenarios. However, the test metrics seem to depend on expert judgment, and the statistics are gathered from manual labor.

Measuring LiDAR performance on real roads is beneficial for capturing a full spectrum of signal characteristics of LiDAR from rich surroundings, compared to the aforementioned artificial test environments. Jokela et al. [8] claimed that previous test methods focused on rain and fog conditions, neglecting other critical weather conditions, such as snow. Therefore, they presented outdoor LiDAR test results in snowy conditions by attaching six LiDAR sensors to the top and front of a test vehicle. However, the test results in an outdoor scene were limited to a qualitative analysis by human experts, such as, for instance, observation of point cloud occlusion by snow. Bijelic et al. [12] released a multi-modal adverse weather dataset covering camera, radar, and LiDAR sensors over about 10,000 km of driving in northern Europe. Based on the multi-modality, they also proposed a sensor fusion network validated on the driving dataset. Dhananjaya et al. [19] proposed an active learning-based weather- and light-level classification model and provided real road datasets for different weather conditions.

In summary, the existing work on the LiDAR performance test in dynamic scenes tends to rely on empirical analysis by human experts, so there is a demand for quantitative and automated test methods.

### 2.3. Lidar Quality Assessment Based on Simulation

Simulation in a virtual environment has clear strengths in validation of automotive sensors, as well as for ADAS/AD functional safety, since it significantly excels at both the sensor-level and the system-level tests by loop testing in combination with modeling of a real-world environment based on test scenarios. Early studies aimed to build a framework for modeling sensors, digitizing the real world, and gathering datasets using driving fleets with attached automotive sensors. Gschwandtner et al. [20] introduced a 3D scan simulator for commercial LiDAR sensors, while Pereira et al. [21] presented an integrated framework for a traffic simulator based on automotive sensor models (LiDAR and camera).

The state-of-the-art literature suggests leveraging physical simulation for LiDAR performance measurement in adverse conditions. Goodin et al. [22] developed a physics-based simulation of the influence of rain on terrestrial LiDAR performance. Manivasagam et al. [23] proposed a LiDAR simulator utilizing both physics-based and learning-based simulations via raycasting over the 3D scene and a neural network for signal deviations. The simulator was capable of performing a repetition test under the test scenarios captured from real self-driving fleets. However, environmental changes were not considered, so that various noise factors from the real world were not fully reflected in the LiDAR sensor model. Hahner et al. [24] simulated LiDAR-based 3D object detection in foggy weather by modeling an attenuation factor driven by fog as a soft target. This model can be applied to an actual LiDAR measurement to evaluate 3D object detection in simulated fog conditions, but their solution is restricted to fog conditions.

Simulation-based methods obviously outperform actual road tests for LiDAR capabilities in adverse weather in terms of quantity, variety, and accessibility. However, there is still a large gap between the real world and the synthesized world. First, the LiDAR industry seeks a dominant mechanism for the LiDAR principles [25]. At this stage, a universal model of attenuation and noise factors is infeasible to create, unlike the current state-of-the-art simulators for standardized sensors, such as front camera [26], and radar [27]. These use an oversimplified LiDAR model in weather simulations, in terms of measurement variance in range and angle, sporadic sensor errors, or multi-level noise factors [28]. Therefore, automotive sensors must be validated on real roads with various test cases, including dynamic driving conditions. We focused on the performance evaluation of real road test cases with a tool to aid in labeling and analyzing the road data.

## 3. Methodology

### 3.1. Test Vehicle

In the Figure 1, we illustrate our test vehicle, which was modified from a consumer-grade SUV to attach automotive LiDAR sensors and accompanying data acquisition systems. We considered the mounting positions of automotive LiDAR sensors for both levels 2, 3 and level 4 self-driving cars, and placed the sensors at three different positions: front, top, and rear. To perform the performance evaluation, the Device Under Test (DUT) was mounted on additional frames. For the front position, we used Valeo Scala gen2 and Innoviz one as the DUT, while, for the top position, we used Ouster OS1 and Velodyne Alpha Prime. For the left and right sides of the rear LiDAR, we considered Velodyne Ultra Puck and Hesai XT32. We changed the combination of LiDAR sensors according to the scope and purpose of each test. In this paper, we focused only on the automotive-graded front LiDAR sensors used for an ADAS feature.

The test vehicle is able to capture not only LiDAR scan data, but the surrounding information. For example, ambient temperature, humidity, acceleration, and illuminance are captured on the cover of the LiDAR sensors using IPETRONIK’s IPElog2 data logger. Additional time series data, such as the position, velocity of the ego-vehicle and video from the windshield camera are also captured and stored in an industrial PC installed in the luggage compartment.

Figure 2 is a schematic diagram of the test vehicle. The major components can be categorized into sensors, signal grabber, data acquisition system, power suppliers, and time-series data analysis. In our schematic, parallel data pipelines, connected to an industrially graded PC, are used to transfer raw image buffers from the front camera, LiDAR point clouds from an Ethernet grabber, and ego-vehicle status into the PC. The feature extraction algorithm in the Section 3.3 was utilized in the time series data analysis for LiDAR point clouds.

### 3.2. Environmental Data Acquisition

This section describes the measurement of the driving environmental data on real roads. The road fleet data was used for two purposes. The first was to observe the resistance of LiDAR performance against extreme driving environments by means of data statistics. Second, we collected road fleet data for a realistic environmental simulation in laboratory conditions. For example, we demonstrate a temperature simulation test result for automotive LiDAR sensors in Section 3.1. In this test, a temperature profile captured in Alaska for low temperature was utilized.

We considered two harsh road environments selected in accordance with our purposes. First, we captured time-series data of road conditions in an Alaskan winter in the USA. The data includes the internal/external temperature, 3-axis position/acceleration, and amount of light coming through the cover of the test LiDAR sensor. Our second series of data were acquired from desert road fleets in a summer in Nevada, the USA. These road data were only collected in conditions under strong sunlight and high-temperature (Figure 3).

For the road data captures, we utilized the data acquisition system mentioned in Section 3.1. This multi-channel data acquisition box collects 1D signals of connected electronic sensors, such as thermocouples, accelerometers, and light meters. An example of the data acquisition process is described in Figure 4. In this figure, a thermocouple is attached on the top of a front LiDAR sensor in order to capture the cover temperature for a road fleet. A capture result is shown as a 2D graph where the *x*-axis measures time and the *y*-axis is the value of an electronic sensor. In this example, we measured the range of the cover temperature of a front LiDAR attached to the front of the test vehicle.

### 3.3. Dynamic Target Feature Extraction Algorithm

Instead of the semantic classification method conducted in a supervised manner, we adopted a density-based clustering approach [29] to collect points of a predefined target from the LiDAR scan data. Deep-learning networks are the most promising candidate for point-wise segmentation via supervision from a large dataset [25]. However, they were not adequate in this study, which dealt with the extraction of LiDAR signals back-scattered from a specific target. First, we investigated the capabilities and signal repeatability of LiDAR systems in harsh weather conditions. In general, weak signals lead to weak inference and erroneous decisions. Second, in this case, the detection of objects was pre-specified before training the models. There was no reason for such a bulky framework for the detection. We had domain knowledge for a test environment surrounding a LiDAR system so we could efficiently design a deterministic algorithm that accounted for it.

The procedure by which we derived target feature extraction algorithm is described in Figure 5. First, we collect scan data labeled with a timestamp from a LiDAR sensor. Second, the scan data is divided into several groups using the DBSCAN clustering algorithm. Third, a spatio-temporal matching algorithm is performed for object tracking. Finally, we obtain segmented points of a target object for the scan data.

The choice of DBSCAN as a scene interpreter was based on the characteristics of LiDAR signals in driving conditions. Point signals from target objects usually have a structural similarity in both spatial and temporal domains. Under this assumption, we performed the spatio-temporal matching between point groups in two consecutive frames. Let $S_{i}^{t}$ be *i*th point groups in a scan frame labeled as *t*. In the clustering selection process, a similarity between $S_{i}^{t}$ and a point group $S_{i}^{t-1}$ in a previous scan frame is measured using the Euclidean distance of the feature vector $f_{i}^{t}=\left(\hat{x},\hat{y},\hat{z},\hat{i},\hat{\theta },\hat{\phi }\right)$ where $\hat{x},\hat{y},\hat{z}$ is the center position of $S_{i}^{t}$ in 3D coordinates, $\hat{i}$ is the mean intensity, and $\hat{\theta },\hat{\phi }$ is the azimuth and elevation angle of the center point. The Euclidean distance between two groups $d(S_{i}^{t},S_{i}^{t-1})$ can be calculated as
(1)$$d\left(S_{i}^{t},S_{i}^{t-1}\right)=∥f_{i}^{t}-f_{i}^{t-1}∥.$$

In Equation (Equation 1), a density difference between a target object and background points becomes more pronounced because we fully leverage the intensity of LiDAR signals in target object segmentation. For all scan frames, the feature extraction algorithm is attractively processed to get segmented points of the target objects. A further optimization for the segmentation quality can be performed using an additional filtering algorithm. In this paper, we used the RANSAC algorithm [30] as a spatial filtering that could reject noise points in $S_{i}^{t}$.

### 3.4. Test Metrics

The dynamic target feature extraction algorithm is able to segment test targets for the LiDAR scan data. In this section, we demonstrate the test metrics that can be acquired from the extracted scan data.

#### 3.4.1. Number of Points

The number of points is counted as the amount of points in the Region of interest (ROI) selected from the algorithm. It indicates the ability to detect an object at a certain distance range, since more scan points on the object with high spatial resolution leads to a better detection rate [2]. It can also be used as an indicator for signal repeatability by observing temporal coherence of the number of scan points in a static test, in which both the DUT and the reference target are fixed.

#### 3.4.2. Intensity

Intensity is, in general, proportional to the signal strength of received laser power reflected from a surrounding object. A modern LiDAR equation models the received laser power with various attenuation factors, such as distance, aperture of photo-diode lens, reflectivity of a target, and laser scattering in air [31]. The received laser power is transformed into electrical signals by the photo diode, and electrical signals are further processed using an unique signal processing module, so the exact definition of intensity varies according to the type of laser receiver integrated into the system and how it is processed through the ECU. A large volume of literature deals with reflectivity estimation for real road objects from point intensity [32]. In this paper, we utilized intensity of points as a signal strength indicator.

#### 3.4.3. Scan Frequency

A definition of the scan frame depends on its mechanism, but for an automotive LiDAR sensor it is generally referred to as a time gap between two consecutive scan frames sent to a client. Scan frequency is especially important for self-driving features since it is highly correlated with a design of system latency and safety validation.

#### 3.4.4. Field of View and Angular Resolution

Angular resolution is the minimum deviation of two distinguishable objects in radial coordinates. It is crucial for long range detection, since the projection area of the laser scanning range becomes exponentially large depending on distance from the origin.

The field of view (FoV) of a LiDAR sensor is defined as the angle between two scan points at the end of the scan range. The FoV is the most fundamental system factor to design in both a sensor system and the ODD of a self-driving car. According to the objective, the specifications on FoV and angular resolution are determined by considering a trade-off relationship between them.

#### 3.4.5. Number of Noise Points

A modern LiDAR sensor has various error sources. In this paper, we refer to the number of noise points as the amount of external noise in point clouds from a pre-described interference source, such as sunlight, laser coming from other LiDAR sensors, back-scattering by rain/fog, multi-reflection of shiny materials, and white noise due to high/low temperature. The noise point can be classified by comparing a test scene, in which external noise sources exist, with a reference scene, in which a targeted external noise source is blocked. In this paper, we distinguished noise points using a similarity measure of each point in scanned point clouds between a test scene and a reference scene.

#### 3.4.6. Range Accuracy/Precision

Range accuracy and precision measures the quality of range detection in scanning of LiDAR sensors. In the literature and in field guidance [1,33], accuracy is calculated as a difference between a reference measurement of the distance of a specified object and the distance measured by the DUT. For this, a high-precision rangefinder is required to obtain reference range data against the DUT. The range precision is a deviation of the measured distance in LiDAR scanning. It can be calculated with a static target within a certain distance of the DUT. Accumulation of scan points leads to statistics for the range accuracy and precision measurement.

## 4. Results

In this section, we demonstrate test scenarios to examine the degeneration of LiDAR performance under 8 different environmental factors. They were performed with our test vehicle, described in Section 3.1. Table 1 describes the test scenario and provides a brief summary on the test results.

There were two types of test scenarios in our study. The first type involved simulating harsh driving conditions in North America in a laboratory setting. These simulations considered factors such as high/low temperatures, direct sunlight on the cover of the DUT, and cover contamination in off-road driving. To test for high temperatures, we measured the ambient temperature at the cover of front LiDAR sensors in the summer in Nevada, the USA (Section 3.2). We configured the minimum and maximum temperatures in the temperature profile and observed the impact of high ambient temperatures on the performance of the front LiDAR sensor in a chamber capable of containing a car. Similarly, we conducted a low temperature test with a captured temperature profile from a winter in Alaska (Figure 4). For the sunlight test, we evaluated the performance of the front LiDAR sensor under the maximum light intensity measured in the Nevada desert using an artificial solar light source. During this test, we evaluated the signal intensity and number of points of a reference object in a static environment while the illumination varied from 0 to 80,000 lux. The cover contamination test was derived from off-road driving in the Nevada desert. We captured the visibility of the DUT every day while driving on a muddy road. In this test, we intentionally restricted the visibility of the DUT by spreading mud on the cover. Similar to the temperature test, we measured the intensity and number of points of a reference object.

The second type of test scenario involved conducting driving tests for an automotive graded LiDAR sensor under various environmental conditions. We examined the position and intensity variations of a front car as it moved on a country road, interference noise from a front LiDAR on a vehicle coming from the opposite side, signal variations during day and night transitions, and front vehicle colors. In the vibration test, we recorded the force applied to the DUT and the signal variations of the DUT using the data acquisition toolbox in Section 3.2. Interference from the opposite LiDAR sensor was observed in a driving test in which a LiDAR-attached vehicle was moved from 0 to 20 m in front of the ego-vehicle. Signal variations during day/night transitions and different front vehicle colors were captured on a straight road as a front car moved forward from 0 to 100 m under various environmental conditions. The signal difference was analyzed to determine the effects of changes in illumination and the color of the front vehicle.

In this paper, the aforementioned analysis was bounded to the Valeo Scala gen2, which is an automotive-graded scanning LiDAR sensor using a 905 nm pulse laser.

### 4.1. Vibration Test

In the vibration test, we drove the ego-vehicle on a country road in Nevada while also driving a reference car at a constant speed in front of the ego-vehicle for performance analysis. We used our feature extraction algorithm (Section 3.3) to segment the points from the front car. Figure 6 shows the coordinates of the DUT and the example scan data, as well as the segmented point for the front vehicle.

To analyze how the impact on the DUT affected sensor signals, we analyzed the standard deviation of the center position of the front car. Figure 7 shows the mean position of the points segmented for the front car in xyz coordinates and their mean intensities, with the *x*-axis indicating the scan frame. The graph indicates that the signal influence was particularly significant for the measured height of an object since the z-axis standard deviation for the center position of the front vehicle drifted by 0.24 m, and the corresponding mean intensity also changed with a deviation of 0.18. Regarding the z-axis in the up direction, such a large drift due to vibration might cause an instant drop in the detection rate of surrounding vehicles.

### 4.2. String Sunlight Test

Sunlight contains the wavelength bands of commercial LiDAR sensors at both 905 nm and 1550 nm, making it a noise source that influences the sensor’s SNR and echo reception rate. Therefore, we analyzed the signal strength reduction rate based on the brightness of sunlight using the HMI DXS’s solar light emulator. In this experiment, the solar emulator was installed in front of the DUT, and the DUT captured points from a reference object attached to the bottom of the emulator. We then analyzed the point data extracted using the feature extraction algorithm. The analysis (shown in Figure 8) revealed that the average intensity decreased by 47%, from 3.5 to 1.84, at 80,000 lux compared to the zero sunlight condition. Meanwhile, the number of points on the reference target only decreased by 1.6%, from 80,656 to 79,310. We chose the maximum illumination of the sunlight simulation as the maximum sunlight in a summer desert in Nevada, the USA. Considering this simulation was conducted in indoor conditions, where direct sunlight was lit in front of the DUT, the maximum detection range and the probability of detection of point clouds could be affected by the sunlight, due to increase in the noise levels of received signals.

### 4.3. High/Low Ambient Temperature Test

Ensuring the durability of automotive sensors against ambient temperature variations is crucial. To this end, we conducted a temperature variation test, simulating extreme temperature conditions based on real road fleet temperature profiles. We used a temperature chamber that can hold a car to control the ambient temperature within the minimum and maximum values recorded. In front of the ego-vehicle, we placed a 1.5 m × 1.5 m square target with 50% reflectivity as a reference object. We gradually increased or decreased the temperature in 5 °C intervals, to simulate the temperature variation experienced in the summer desert and winter arctic regions.

The results, as shown in Figure 9, indicate that the intensity of the reference target remained almost the same as the initial temperature in both high and low temperature tests. For instance, in the low-temperature test (Figure 10), it only decreased by 1.3% from 0 °C to −30 °C. Additionally, we observed no significant change in the number of points against temperature variations on both sides. This outcome suggests that automotive-grade LiDAR sensors may have a compensation system for temperature variations, since the range characteristics (accuracy, precision, and detection rate) are highly relevant to the temperature of the LiDAR ranging module [34].

### 4.4. Interference by External Laser Source

Preventing interference from external laser sources, especially from other LiDAR sensors in oncoming vehicles, is critical to ensure the accurate detection of surrounding objects. To evaluate the anti-interference system of LiDAR sensors, we conducted a test in which a LiDAR-equipped vehicle moved from 0 to 20 m in front of our ego-vehicle. We measured the number of noise points generated by the LiDAR sensor mounted on the oncoming vehicle. To isolate the noise, we captured a reference scene with no interference in advance and then captured point clouds from the DUT while driving towards the oncoming vehicle, and, finally, we segmented the noise points using structural similarity comparison.

As shown in Figure 11, we observed a significant reduction in interference noise from the LiDAR sensor in a short distance of approximately 3 m. At a distance of 1 m, a large amount of interference noise (18,111 points with a mean intensity of 1.36) was present, but this reduced to 467 points at 3 m distance. This result demonstrates the effectiveness of the LiDAR sensor’s anti-interference system in mitigating noise caused by external laser sources.

### 4.5. Cover Contamination Test

In real-world driving conditions, automotive sensors can be obstructed by various types of contaminants, such as dust, snow, rain, or bugs. Such blockages can cause ADAS features to malfunction, making it necessary to develop better coating or cleaning systems to prevent them. In this test, we simulated contamination on the covers of LiDAR sensors as would occur in muddy off-road conditions in Nevada, the USA. Prior to the simulation, we measured the visibility of a front LiDAR sensor immediately after driving in muddy off-road conditions. During the off-road trials, we drove the test vehicle in harsh conditions having numerous puddles, causing mud to cover the front LiDAR sensor. After just one day of driving, the visibility of the sensor was nearly zero.

For the contamination simulation, we applied mud to the DUT until its visibility matched the measured visibility. For each trial, we measured the intensity and number of points of a 1.5 m × 1.5 m square target (50% reflectivity).

The results, shown in Figure 12, indicate that when the cover was fully covered by mud, the intensity decreased by up to 95.1% and the number of points decreased by up to 99.9% compared to the normal operation of the DUT. It is important to note that these results do not imply that automotive LiDAR sensors are weak on muddy roads. Rather, this test intended to observe the behavior of the sensors in pure blockage situations and to develop a fault mode operation to deal with blockages.

### 4.6. Color Change Test

Due to eye safety regulations for laser-based sensors, the power of induced laser for automotive LiDAR sensors is limited, leading to a restricted detection range. In such cases, the surface reflectivity of the induced laser wavelength is a crucial factor in detecting surrounding objects. Therefore, it is imperative to consider color variations of commercial vehicles in the detection process of automotive LiDAR sensors.

For this test we drove two identical model cars (one black and one white) on a straight road in Nevada, the USA, at night. We captured points from the front car as it moved from 0 m to 100 m in front of the ego-vehicle at a constant speed of 20 km per hour. In each scan frame, the feature extraction algorithm in Section 3.3 was used to segment the points of the front car.

The results in Figure 13 indicate that the overall intensity of the white car was higher than that of the black car. For example, for a short and middle range (0 to 80 m), the intensity curve was 10% higher than that of the black car. In the long range (above 80 m) the intensity difference converged to near 0.

While there was a clear intensity change with respect to the color of the front car, there was no significant variation in the number of points with respect to distance. We observed that the black and white cars had almost the same number of points for all distances.

Overall results indicate that the detection quality for the surrounding cars would not be changed according to the color of the car, in terms of probability on detection, since there was no significant difference of number of points captured from the moving car for all distances. Nevertheless we observed intensity difference. This fact did not lead to decrease in the true positive rate.

### 4.7. Day and Night Transition Test

As discussed in Section 4.2, natural sunlight can act as an external noise source and affect the scan data of the LiDAR sensor. To investigate this effect, we conducted tests, similar to the color change case (Section 4.6), in both day and night conditions. In this test, we used a white car as the front car and varied the time of day during which the test was conducted.

The results in Figure 14 show that the intensity dropped similarly for both day and night conditions as the distance increased. However, the rate at which the number of points dropped was noticeably different. Under night conditions, more points were captured at intervals of 10 m to 50 m compared to day conditions. This observation suggests that natural sunlight can degrade the signal strength and SNR of a LiDAR sensor.

## 5. Conclusions

In this paper, we presented a comprehensive test framework to evaluate the performance of LiDAR sensors under harsh weather conditions. Our approach included a specially equipped test vehicle, data acquisition systems for three different data pipelines, and a feature extraction method for qualitative analysis of LiDAR point clouds. Our method was designed to analyze dynamic test cases where the ego-vehicle or test targets were in motion. In the vibration test, our feature extraction model successfully extracted ROIs of targets while the ego-vehicle and front car were driving on a country road.

Using our feature extraction model, we conducted two types of performance tests: environmental simulations, based on real road fleet data in extreme hot and cold weather conditions, and dynamic field tests on various real roads. Our test results clearly demonstrated that environmental factors, as described in Section 4, can significantly affect the performance of LiDAR sensors.

In the future, we plan to extend our framework to real-time scene interpretation from point clouds to obtain more comprehensive information for LiDAR sensor performance analysis. This extension will be useful for multi-modal analysis of LiDAR capabilities for real road fleets. We plan to adopt a deep learning scheme for multi-object segmentation [35] in this framework.

## Figures and Tables

**Figure 1 sensors-23-03892-f001:**
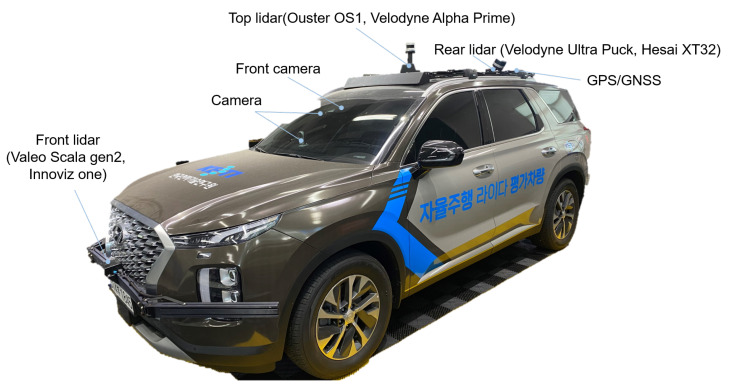
LiDAR test vehicle. Test devices are attached on Front, Top, and Rear positions using additional metal structures. At each position, the corresponding LiDAR sensors can be replaced and mounted as the DUT.

**Figure 2 sensors-23-03892-f002:**
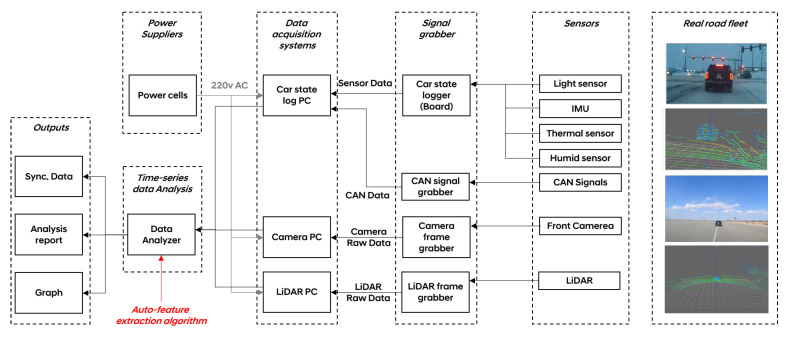
Schematic diagram of the test vehicle. In real road fleets, the test vehicle collects front camera images, point clouds of LiDAR sensors, and time-series signals of environments and car status. The gathered data is stored in industrial PCs installed in the luggage compartment. Additional batteries are used to supply electronic power to each component.

**Figure 3 sensors-23-03892-f003:**
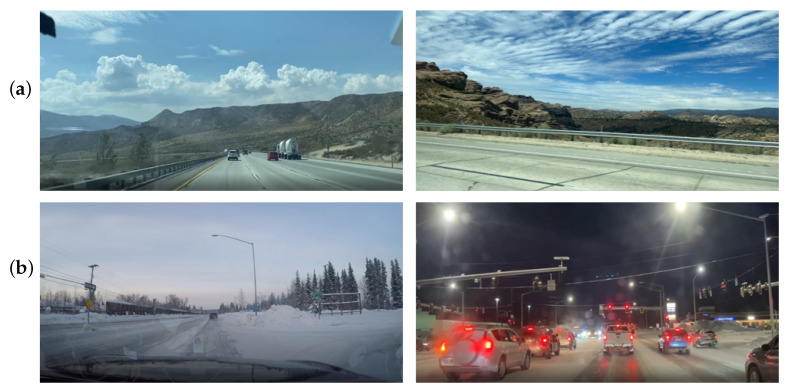
Real road fleet examples for the collecting of driving data (**a**) under strong sunlight and high temperature (above 40 °C) in Nevada, the USA (**b**) under low sunlight and low temperature (below −17 °C) in Alaska, the USA.

**Figure 4 sensors-23-03892-f004:**
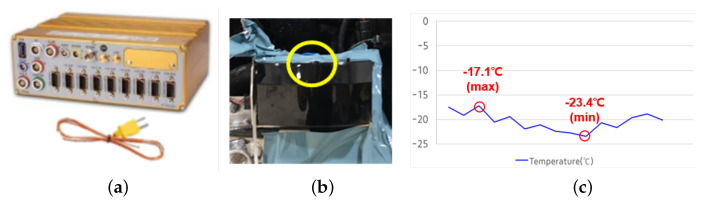
Car state logger example (**a**) IPETRONIK’s IPElog2 as a data acquisition toolbox. (**b**) Cover temperature measurement setup using a thermocouple. (**c**) Measured temperature profile in Alaska. The yellow mark indicates the attachment location of a wired thermal sensor connected to the IPElog2 (shown in (**a**)).

**Figure 5 sensors-23-03892-f005:**
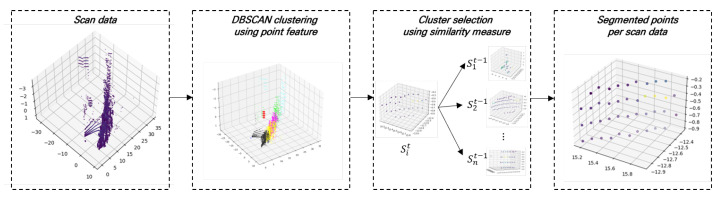
Overall procedure of feature extraction algorithm.

**Figure 6 sensors-23-03892-f006:**
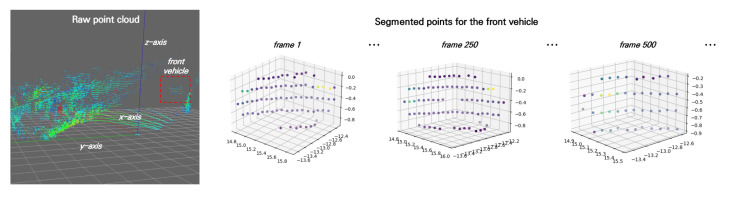
Example of scan data in the vibration test. (**left**) raw point cloud in a scan frame, (**right**) segmented points of the front vehicle using the feature extraction algorithm.

**Figure 7 sensors-23-03892-f007:**
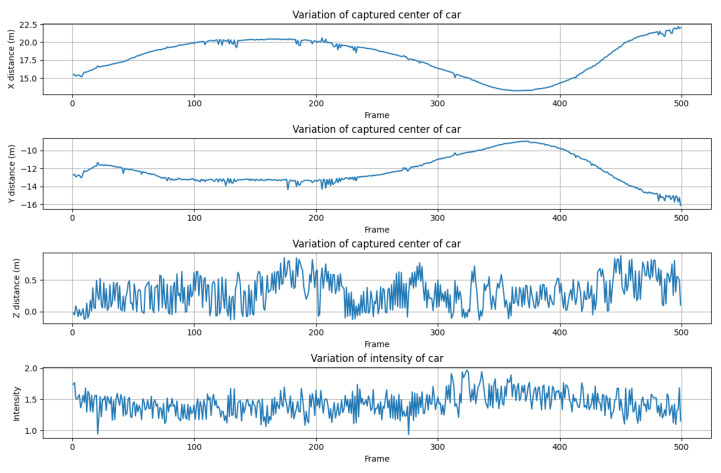
Example of vibration test result. We observed the position and the intensity of the front car being driven at a constant speed (30 km/h) on a country road.

**Figure 8 sensors-23-03892-f008:**
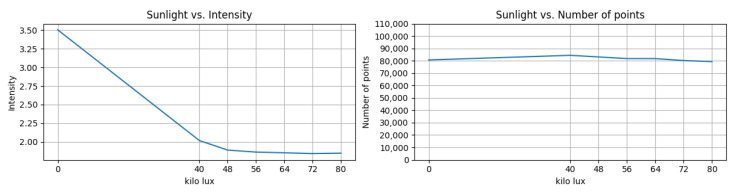
Example of strong sunlight test. The number of points in the reference object and the mean intensity vs. amount of illumination (lux) was plotted in this figure.

**Figure 9 sensors-23-03892-f009:**
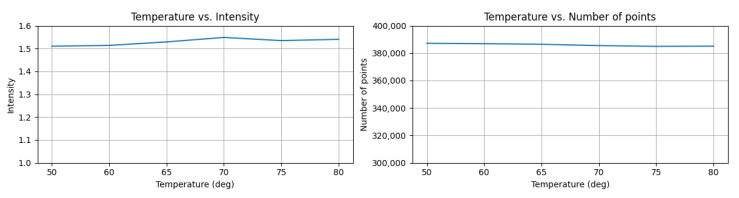
Example of high temperature test results. (**Left**) mean intensity versus ambient temperature (**Right**) number of points versus ambient temperature.

**Figure 10 sensors-23-03892-f010:**
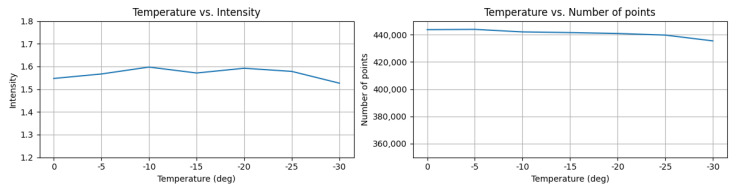
Example of low temperature test results. (**Left**) mean intensity versus ambient temperature (**Right**) number of points versus ambient temperature.

**Figure 11 sensors-23-03892-f011:**
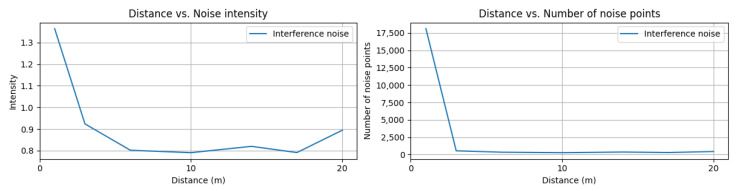
(**Left**) mean intensity of interference noise from a LiDAR sensor faced with the DUT (**Right**) number of LiDAR interference noise per distance.

**Figure 12 sensors-23-03892-f012:**
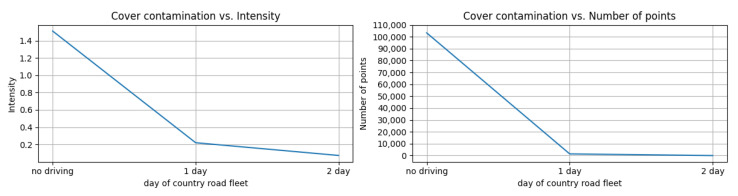
(**Left**) mean intensity of captured points from a reference target (**Right**) number of captured points from a reference target. For each graph the *x*-axis indicates the contamination condition captured by an off-road fleet in Nevada.

**Figure 13 sensors-23-03892-f013:**
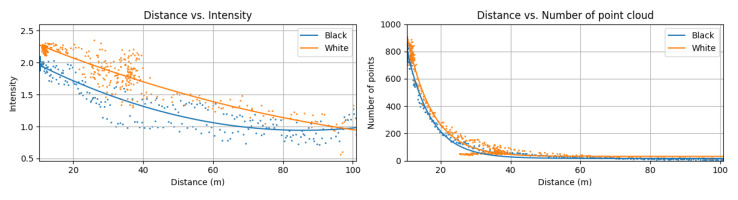
(**Left**) mean intensity of captured points from black/white car according to the distance (**Right**) number of captured points from black/white car according to the distance.

**Figure 14 sensors-23-03892-f014:**
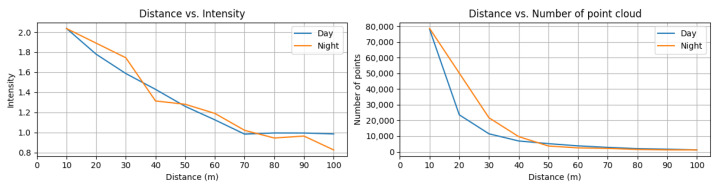
(**Left**) mean intensity of captured points according to the distance in day/night (**Right**) number of captured points according to the distance in day/night.

**Table 1 sensors-23-03892-t001:** Summary of environmental test scenarios and results.

Type		Environmental Conditions	Test Results
Simulation with road fleet data (Section 3.2)	Cover contamination	Mud was spread mud on the cover of the DUT based on visibility of a front LiDAR sensor from an off-road fleet. A reference target was placed in front of the DUT at a 5 m distance.	The signal intensity and the number of points decreased when the cover of the LiDAR was obscured by mud contamination.
	Strong sunlight	Applying direct sunlight to the DUT while the DUT captured points from a reference target. Illumination flux was controlled by measured maximum sunlight on a desert surface in Nevada	The mean intensity decreased due to sunlight but the number of points from a reference target remained the same. The intensity drop rate was 47% at the highest illumination flux compared to the zero sunlight case.
	High temperature	Simulating ambient temperature using a large temperature chamber capable of containing a test car, with temperature profiles captured from an extremely hot area in Nevada, the USA	As the temperature increased, the signal strength increased by 19.8% compared to the initial stage.
	Low temperature	Simulating ambient temperature using a large temperature chamber capable of containing a test car with temperature profiles captured from an extremely cold area in Alaska, the USA	At low temperatures, the difference of signal intensity changed insignificantly.
Performance test in dynamic conditions	Vibration	Recording the force applied to the DUT and the signal variations of the DUT using the data acquisition toolbox and driving feature extraction algorithm (Section 3.3)	Position in z-axis and intensity of a front vehicle drifted highly in response to road vibrations. A high deviation in the z-axis position and intensity were measured.
	Interference	Capturing interference noise from a front LiDAR on a vehicle coming from the opposite side while the opposite car was being moved from 0 to 20 m distance to the ego-vehicle	Interference noise might occur at a very short distance (e.g., 1 m). It drastically decreased as the distance to the interference source increased.
	Color change of front car (reflectivity of a target)	Driving the front car in a straight road from 0 to 100 m distance to the ego-vehicle. We utilized the black and white car which had a different reflectivity to 905 nm wavelength of light.	A 33.2% drop rate was measured in the intensity curve. There was no significant change in the number of points in the graph.
	Day and night transition	Driving the front car on a straight road from 0 to 100 m distance to the ego-vehicle in day and night conditions	The signal intensity decreased under day conditions compared to night conditions.

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
