# Peer review of "An Automotive LiDAR Performance Test Method in Dynamic Driving Conditions"

_sensors, 2023, doi:10.3390/s23083892_

Round 1

Reviewer 1 Report

the paper addresses an interesting topic and it is very relevant for autonomous driving. However I find a lack of explanations, analysis and discussion in the paper. 

I will start with some small remarks like missing reference at line 246 or typo at line 419.

I don't understand the description of the vehicle with an enumeration of lidar sensors but the analysis doesn't mention anything about which lidar model was used for each experiment. 

Graph in Figure 6 doesn't mentioned anything about how is the reference frame oriented with respect to the moving vehicle. 

What I really and most important miss in the paper is a good discussion about the results obtained during the experiments.

Why the curve in Figure 6 is like it is?

In Figure 12, it is not what most of the readers expect to see. It is unexpected that the black car has larger intensity values than a white car at 80-100 meters distance. What is the orientation of the target pattern with respect to the lidar? is the orientation influencing more into the black? or the white car?

Concerning references, I think a reference to "LIBRE: The Multiple 3D LiDAR Dataset" would be a good reference to add. It makes a vert good comparison between lidars and also extended experiments with them. 

Author Response

We really appreciate your valuable review and comments. In order to address each point you raised in the first draft, we would like to divide our responses by topic as attachment.

Reviewer 2 Report

This paper proposes a test framework for LiDAR performance under harsh weather conditions. The work is interesting and significant. The below suggestions may help improve the quality of the paper.

     1.     The abstract requires more descriptions about methodologies.

2.     The contributions should be improved deeply to show the novelty. Current version is not attractive enough.

3.     Deeper discussions are encouraged in the result part to show more interesting findings.

4.     Writing should be improved. 

Author Response

(The authors gave the same response as above.)

Round 2

Reviewer 1 Report

I still think the results need to be further analyzed and discussed. The unexpected result of a black car having hiher intesnity values than a white car needs to be analized in detail to find the reason of the results. I don't accept the results without any further investigation. If more experiments are required, then the paper should include them. 

Author Response

We really appreciate your comment pointing out unexpected results on the color change test. Regarding the comment, we have investigated the result and reply the comment in attachments. 
